# Nutraceuticals: A New Challenge against Cadmium-Induced Testicular Injury

**DOI:** 10.3390/nu14030663

**Published:** 2022-02-04

**Authors:** Herbert Ryan Marini, Antonio Micali, Giovanni Squadrito, Domenico Puzzolo, José Freni, Pietro Antonuccio, Letteria Minutoli

**Affiliations:** 1Department of Clinical and Experimental Medicine, University of Messina, 98125 Messina, Italy; hrmarini@unime.it (H.R.M.); gsquadrito@unime.it (G.S.); lminutoli@unime.it (L.M.); 2Department of Human Pathology of Adult and Childhood, University of Messina, 98125 Messina, Italy; amicali@unime.it (A.M.); pietro.antonuccio@unime.it (P.A.); 3Department of Biomedical and Dental Sciences and Morphofunctional Imaging, University of Messina, 98125 Messina, Italy; puzzolo@unime.it

**Keywords:** cadmium, testis, oxidative stress, inflammation, apoptosis, blood–testis barrier, infertility, nutraceuticals, Mediterranean diet

## Abstract

Cadmium (Cd) is a widespread heavy metal and a ubiquitous environmental toxicant. For the general population, the principal causes of Cd exposure are cigarette smoking, air pollution and contaminated water and food consumption, whereas occupational exposure usually involves humans working in mines or manufacturing batteries and pigments that utilize Cd. The aim of the present review is to evaluate recent data regarding the mechanisms of Cd-induced testicular structural and functional damages and the state of the art of the therapeutic approaches. Additionally, as the current literature demonstrates convincing associations between diet, food components and men’s sexual health, a coherent nutraceutical supplementation may be a new valid therapeutic strategy for both the prevention and alleviation of Cd-induced testicular injury. The toxic effects on testes induced by Cd include many specific mechanisms, such as oxidative stress, inflammation and apoptosis. As no specific therapy for the prevention or treatment of the morbidity and mortality associated with Cd exposure is available, the development of new therapeutic agents is requested. Dietary strategies and the use of nutraceuticals, particularly abundant in fresh fruits, beans, vegetables and grains, typical of the Mediterranean diet, are recommended against Cd-induced testicular injury.

## 1. Cadmium: The Beginning of History in Humans

In agriculture and industry, a large quantity of pesticides and chemicals, necessary for the increase in the economic yield of soil and the production of electronic devices, vehicles, paints and other materials, are released into the environment [1,2].

Regarding the heavy metals, some (iron (Fe), cobalt, copper (Cu), manganese, zinc (Zn), and molybdenum) are essential at low concentrations for metabolic activity [3]. Other, in particular, mercury (Hg), cadmium (Cd), nickel (Ni), arsenic and lead (Pb), are not biodegradable and have a long biological half-life [4]; thus, they accumulate in the environment and are then absorbed by the human body. As they have no identified useful biological function, they are considered as “toxic heavy metals” (THM) [5].

Once in the body, THM affect different physiological processes, even at very low concentrations [6], as they are used as a substitute of essential elements: Pb can take the place of calcium and Cd the place of Zn [7]. As a result, they can cause numerous diseases, such as kidney toxicity [8], impaired lung function [9], neurotoxic effects [10,11], hormonal imbalance [12], abnormal spermatogenesis and infertility [13], and even cancer [14,15,16]. Furthermore, THM can cross the placenta and interfere with fetal developmental processes [17].

Among THM, Cd is a major heavy metal toxic to the environment that causes harmful events in many biological processes of humans, animals and plants [18]. Cd is a physically soft metal, not biodegradable, and with a half-life of 15–30 years in humans [13]. In the industry, it has a wide range of applications owing to its outstanding characteristics against corrosion and resistance to oxidation: in fact, it is used in the construction of Ni-Cd batteries, in the production of many materials as color element, in the coating of mechanical parts and in photography [18]. Among the different compounds, only a small part (1–10%) of the orally ingested dose of cadmium chloride (CdCl_2_) is absorbed through the intestine [19,20]; instead, cadmium oxide (CdO) is absorbed through respiration.

In humans, Cd exposure is mainly due to cigarette smoke, as it is accumulated in tobacco plants with a concentration ranging from 650 to 3630 ng/g tobacco [21]. From the cigarette smoke 50–100% of the small particles of Cd are absorbed from the alveolar epithelium, pass through the pulmonary interstitial tissue and enter the circulation [22]. Environmental respiratory intake is particularly important in workers of mines, and paint and battery factories, where dust and fumes may contain significant amounts of Cd. Then, toxicants, once passed the upper airways protection structures and entered into the alveoli, are absorbed and transferred in the blood [23].

In nonsmokers, Cd exposure is usually linked to food intake, with different degrees of absorption based on the type of the toxicant, its dose and the frequency of exposure [24]. The efficiency of Cd absorption from the gastro-intestinal tract is low; in humans the uptake is about 3–7% [25]. As to the levels of Cd consumed with food, daily human intakes range from 10 μg Cd/kg body weight, considered as high exposure, to 0.1 μg/kg, but should not exceed 21.4 μg [26]. As a matter of fact, some discrepancies are present in different countries, related to the industrialization level, type of job and food habits of the local culture. In fact, in China, cereals were the most significant contributors to the dietary intake of Cd, whereas in France and Spain, the highest concentration of Cd was in fish and fish products [27]. In the USA, the food groups that contributed most to Cd intake were cereals and bread, followed by leafy vegetables and potatoes [28]. In Sicily, fish, vegetables and potatoes were particularly abundant in Cd levels [29,30].

Once in circulation, about 90% of Cd is bound to serum albumin and α2-macroglobulin [31]. In this way, Cd is carried to the liver, where the complex is degraded; as a result, metallothioneins (MT), small cysteine-rich proteins, are synthetized [32]. Even if the four main isoforms (MT-1 to MT-4) are known, the administration of heavy metals, such as Cd, causes the biosynthesis of MT-1 and MT-2 acting on specific transcriptional factors; on the contrary, no specific role in the detoxification of heavy metals is known for MT-3 and MT-4 [33]. MT-1 and MT-2 may bind both functional (Cu, Zn, Fe) and toxic (Cd, Hg, Ni, Pb) heavy metals through four thiol groups of cysteines. When bound to Cd, MTs are stored in hepatocytes as Cd-MT complexes, thus trying to protect the cells from toxic Cd ions. Even if large amounts of these complexes are accumulated in the liver, small quantities are discharged into the blood, filtered by the glomeruli and absorbed by renal tubular cells [34]. Once in the cells, the Cd-MT complex is degraded by lysosomes to amino acids and free Cd ions. Finally, when compared with other heavy metals, Cd shows low toxic effects to the fetus and the central nervous system [34].

## 2. Nutraceuticals: The New Frontier of Nutrition

The word nutraceutical, made from the terms “nutrient” and “pharmaceutical”, was coined by Stephen DeFelice in 1995, who defined nutraceuticals as “foods (or part of a food) that provide medical or health benefits, including the prevention and/or treatment of a disease” [35]. Currently, the term “nutraceutical” applies to a wide range of products, such as dietary supplements, herbal/botanical products, specific processed foods (functional foods) and also isolated nutrients. The terms “dietary supplements” and “functional foods” are used without distinction as synonyms, although there are substantial differences between them; we then classify: (i) “dietary supplements”, (ii) “functional foods” and (iii) “nutraceuticals” [36].

Many new nutraceuticals have been recently added and include phenolic compounds, organic acis, tocopherol, carotenoids, anthraquinones, terpenes, alkaloids, isothiocyanates and mono- and poly- unsaturated fatty acids, among others.

Indeed, consumers in different parts of world demonstrate great acceptance of these products because are particularly attracted by their plant origin, and also nutraceutical-producing companies, beyond the nutritional value, are strongly interested by the growing market value of nutraceuticals that represent, to date, a therapeutic alternative to pharmaceuticals [37].

## 3. Cadmium and Toxicity of Testis: Potential Molecular Targets of Nutraceuticals

In populations living in Cd-polluted environments, higher numbers of testicular cancer cases [38], reduced semen quality, even infertility [39] and delayed puberty with reduced testicular growth [29], were observed, although with differences in susceptibility referred to genetic polymorphism [38].

An important reduction in male fertility has been associated with occupational exposure to Cd [40], whereas varicocele and other reproductive disorders account for only 23% of damaged reproductive conditions [4]. As a consequence of Cd exposure, severe hemorrhage, edema, necrosis of Leydig cells and decreased testosterone concentrations in the plasma and testes, reduced junctions of germ cells in the seminiferous tubules and a loss of integral membrane proteins at the Sertoli cell interface of the blood–testis barrier (BTB), a reduction in the count and motility of sperm have been observed [41,42,43,44,45,46,47]. Furthermore, Cd is also classified as a potential human carcinogen by the International Agency for Research on Cancer [48]: in fact, Leydig cells death or even cancer were demonstrated [49].

Even if the causes of Cd-induced reproductive toxicity have been not yet fully elucidated [50], several mechanisms have been demonstrated, such as oxidative stress [34,51], DNA changes [52], apoptosis [53], inflammation [54], disruption of the BTB [45,46,55] and hormonal imbalance [44,56].

In experimental models, we used a single dose of CdCl2, that is 2 mg/kg/day, after having tested the effects of different doses able to induce testis damages [44]: this point deserves a specific discussion. In fact, this dosage is in accordance with previous studies on the topic [57,58] and was used to evaluate the toxicity of Cd without causing extensive and irreversible damages of the seminiferous epithelium. Different doses, either lower [59] or higher [60,61], different methods of administration, oral [62] or subcutaneous [63], and different durations of Cd administration [64] are able to induce structural changes in the male genital apparatus. In addition, it must be considered that exposure to heavy metals such as Cd is different from environmental conditions typical of polluted areas [29,56,65], where many substances can contribute to amplify the toxic action of Cd. In any case, the results obtained with a single dose of Cd allow us to attribute all biochemical and morphological changes observed in experimental conditions to Cd.

Specifically, in this context, our research lab and other groups tried to identify the putative pathophysiological mechanisms involved in Cd toxicity, such as oxidative stress, inflammation, apoptosis, hormonal imbalance, that, in turn, through an intriguing molecular crosstalk, are all able to induce severe BTB damage. Based on these findings, new possible therapeutic approaches involving the relationship between dietary strategies, use of nutraceuticals, nanostructures and nanotechnologies are recommended against Cd-induced testicular injury.

So far, recent evidence and the experience of our research lab suggested that nutraceuticals can be added easily and affordably to the daily diet and are expected to have very few side effects compared with chelation therapy. As a matter of fact, observational studies showed a positive association between sperm parameters and following a Mediterranean diet [66,67], possibly correlated to the antioxidant characteristic of the different nutrients typical of this dietary approach [68], able to positively impact on semen parameters [69,70,71]. Then, in human exposition to heavy metals, such as Cd, a coherent use of nutraceuticals may be a new reliable strategy, particularly in organs which are very susceptible to damage, such as testis, suggesting that the recent results reported in the present review could be translated in clinical practice (Figure 1). Then, in light of this background, we conducted a narrative review including experimental and clinical studies with the aim to better define the intriguing relationship between Cd, testis and nutraceuticals. Of course, we hope that, in upcoming years, through the development of knowledge about this controversial topic it will be possible to assess a more detailed systematic review with the main goal to estimate the effectiveness of nutraceutical intervention on Cd-induced testicular injury.

## 4. Cadmium-Induced Oxidative Stress Causes Testicular Injury and Represents an Attractive Molecular Target for Nutraceuticals

Oxidative stress is one of the main mechanisms of Cd-induced testicular damage [72]. In fact, Cd promotes the production of reactive oxygen species (ROS), inducing sperm DNA damage [73]. ROS are particularly important in Cd toxicology, either in vitro [74] or in vivo through all routes of exposure [75].

Under physiological conditions (oxidative eustress), ROS show positive effects, as they control several physiological responses [76]. In fact, ROS regulate the immune response and vital cellular functions, such as growth, differentiation and migration, control intercellular junctions, trigger different pathways, such as Nuclear Factor-κB (NF-κB) and mitogen-activated protein kinases (MAPKs), and modulate apoptosis [77].

Instead, an increased or persistent production of ROS induces harmful effects, leading to cell dysfunction or death. The mechanisms involved in Cd toxicity include the reduction in glutathione (GSH) and protein-bound sulfhydryl groups, causing an increased production of ROS, such as hydroxyl radicals, hydrogen peroxide and superoxide ions, able to induce lipid peroxidation [78]. It must be kept in mind that spermatozoa are extremely sensitive to ROS, owing to their high amount of polyunsaturated fatty acids and their low capacity to repair DNA [23,73]. As a result, a ROS increase is able to induce damage in the key enzymes involved in testicular steroidogenesis, with a reduction in testosterone levels [79], and in the process of spermatogenesis [80], with abnormal sperm morphology and decreased sperm number, finally inducing male infertility.

Oxidative stress seems to be crucial in the etiology of Cd-induced male reproductive toxicity in humans and animals. Thus, antioxidant therapy is considered an important approach for the intervention of Cd toxicity.

Alpha lipoic acid (ALA) is made naturally in the body and may protect against cell damage in different conditions. Spinach, broccoli and yeast are particularly rich in alpha lipoic acid. ALA, the “universal oxidant,” has been used for decades in Europe, especially Germany, to treat nerve conditions, including nerve damage resulting from poorly controlled diabetes [81]. The exogenous administration of ALA was shown to increase its free levels, thus lessening the oxidative stress both in vitro and in vivo [82]. Moreover, as ALA is able to chelate heavy metals such as Cd [83] when administered to rats chronically treated with Cd, it defends against testis lesions induced by increased levels of ROS, reestablishing the antioxidant status, normalizing steroidogenesis and reducing Cd increase [83].

Hesperetin, a Citrus flavonoid [84], exhibits antioxidant, anti-inflammatory and metal chelating activities. In vitro studies showed a potent radical scavenger role [85]. Accordingly, after the Cd challenge, the supplementation with hesperetin acted positively on the structure of seminiferous tubules, suggesting a protective role against increased ROS levels [86].

The flavonoid quercetin is able to attenuate the toxicity of environmental Cd on reproduction, through a reduction in ROS [87], by restoring the GSH level and enzymatic antioxidant endogenous system (superoxide dismutase (SOD) and GSH-Peroxidase).

Flavocoxid, a flavonoid containing both baicalin and catechin, with antibacterial, antiviral and anticancer properties [44,88], and positive cardiovascular effects [89], protected testis from Cd-induced damage, owing to its direct antioxidant activity, preventing the experimental generation of malondialdehyde (MDA).

In recent years, the analysis of phytomelatonin from several plant species has opened the door to its use as a nutraceutical compound, due to its outstanding actions at the cellular and physiological levels, especially its protective effect in plants exposed to diverse stress situations. Structurally, melatonin and phytomelatonin are the same molecule. Specifically, melatonin is the term first proposed in 2004, and refers to melatonin of plant origin [90]. Several studies have reported on detection of phytomelatonin in a variety of vegetables, fruits, seeds and medicinal herbs, and also in wild plants [91]. To date, the phytomelatonin content of coffee beans is the highest recorded in plant material. Moreover, apple and cherry, Goji berry, tomato and pepper fruits also showed a high phytomelatonin content. Overall, aromatic and medicinal plants had higher levels of phytomelatonin than seeds and fleshy fruits, whereas leaves, stems, seedlings and roots presented higher phytomelatonin content than fruits [80]. Consequently, botanical herbs are optimal candidates for use in the future, including the aim to counteract Cd-induced damage in testes. Melatonin pretreatment significantly lowered Cd toxic effects by inhibiting MDA levels, normalizing GSH and SOD activities, and decreasing the tumor necrosis factor (TNF)-α and interleukin (IL)-1β levels [92], thus protecting against Cd-induced testicular toxicity.

An important role in normal testicular development, spermatogenesis and spermatozoa motility is experimentally played by selenium (Se) in vivo and in humans [93]. Regarding the mechanism involved, Se is an important component of selenoproteins, such as GSH-Px and thioredoxin reductases [94,95]. Moreover, Se might also increase testosterone levels by stimulating synthetic enzyme activity [96].

Finally, among the natural nutraceutical antioxidants, myo-inositol (MI) had a significant role in reducing oxidative stress, as demonstrated by the increased levels of SOD, catalase (CAT) and GSH in juvenile carp [97]. Laboratory studies demonstrated that Cd adversely affects adipose tissue physiopathology through several antioxidant mechanisms, thus contributing to increased insulin resistance and enhancing diabetes [98]; intriguingly, a supplementation with a combination of MI and d-chiroinositol is an effective and safe strategy for improving glycemic control in type 2 diabetes [99,100,101].

We recently demonstrated the protective effect of MI, a minor role of Se and an evident positive role of the association between MI and Se on Cd-induced damages to the testes [46]. MI alone or associated with Se might protect the testes in subjects exposed to toxicants, at least to those with behavior similar to Cd.

## 5. Cadmium Toxicity: Inflammation Is a Significant Molecular Pathway Targeted by Nutraceuticals

Current literature showed that persistent oxidative stress can induce an amplification of inflammation, which is considered a significant pathological pathway in the testis after Cd challenge [54].

A central role in this process is played by cytokines able to control many testicular physiological functions, among which spermatogenesis, hormonal synthesis by Leydig cells, and regulation of the extratubular compartment are included. TNF-α is a crucial cytokine produced by Sertoli and germ cells in the testis; it strengthens and protracts the inflammatory response stimulating the release of cytokines and mediators. In addition to TNF-α, Cd exposure is able to cause an increased concentration of IL-1β, IL-6, IL-10, inducible Nitric Oxide Synthase (iNOS), Nitric Oxide (NO), Cyclooxygenase (COX)-2 and interferon (IFN)-γ owing to the augmented oxidative stress, which can have negative effects on spermiogenesis, cause structural changes in Sertoli cells junctions and interfere with the adhesion of the germinal epithelium to the tubular basement membrane [62].

Another negative impact of Cd is the induction of the NOD-like receptor protein (NLRP) inflammasome. Many inflammasomes have been described [102]. The most fully characterized is the NLRP3 inflammasome because it answers to different chemical and physical stimuli, inducing many diseases when out of control [103]. Once activated, NLRP3 activates caspase-1, which in turn induces the activation of IL-1β and IL-18 from pro-IL-1β and pro-IL-18 [104].

After the Cd challenge, NF-κB is considered to be a detrimental trigger in an inflammatory cascade in the testis. Under normal physiological conditions, NF-κB is bound to its inhibitory subunit, IkB, in the cytoplasm. After negative stimuli, it grows free from its subunit, migrates into the nucleus and induces the transcription of pro-inflammatory genes [105].

Cd-induced testis inflammation resulted in widespread necrosis and vacuolization of the seminiferous epithelium cells, together with interstitial tissue edema and hemorrhage, which was able to induce male infertility [23].

In this regard, it has also been shown that melatonin has an anti-inflammatory effect and suppresses the synthesis of pro-inflammatory cytokines [92].

Additionally, nutraceutical flavocoxid exerts important anti-inflammatory activity, inhibiting COX-2 and 5-Lipoxygenase (LOX), and reducing the synthesis of Prostaglandin (PGE)_2_ and Leukotriene (LT) B4 [106].

Recently, bergamot juice (BJ) has gained growing scientific interest [107,108]. In fact, it has been shown that BJ and its flavonoid-rich fraction (BJe) exert hypolipemic and hypoglycemic activity [109], anticancer [110], anti-infective [111], neuroprotective [112], antioxidant [113] and anti-inflammatory effects [114]. Our research lab recently confirmed the antioxidant effect of BJe, alone, or in combination with curcumin (Cur) and resveratrol (Re) against Cd-induced testicular injury, and we suggested that the bioactive compounds present in the phytocomplex can act in a multitarget mode of action [47].

Adenosine receptors were demonstrated to be a striking target in the management of healing disorders [115,116,117]. In fact, the inflammatory cascade is blocked by A2A receptor activation, thus ameliorating tissue repair and the healing process. In previous papers, we demonstrated the positive role of an A2A receptor agonist, polydeoxyribonucleotide (PDRN), a biologic drug obtained from the gonads of trout and containing different polynucleotides, in reducing inflammation and improving tissue repair [118,119,120,121]. In fact, when PDRN was administered to Cd-challenged mice, it lowered phosphorylated extracellular signal-regulated kinase (pERK) 1/2 expression, indicating the possibility that mitogen-activated protein (MAP) kinase could represent a trustworthy molecular target for nutraceuticals.

Accordingly, our research lab indicated that MI alone or associated with Se significantly reduced iNOS and TNF-α expression, indicating the significance of molecular crosstalk between oxidative stress and inflammatory cascades in the pathophysiology of Cd-induced testis damage [46].

## 6. Nutraceuticals and Cadmium-Induced Apoptosis in Testis

Oxidative stress and inflammation are also able to induce apoptosis in vitro [122] and in vivo [123].

Apoptosis is a programmed cell death able to cause different biochemical and morphological events, such as DNA fragmentation, chromatin condensation, cell shrinkage, membrane blebbing and the appearance of apoptotic bodies [4]. In the seminiferous tubules, during the normal process of spermatogenesis, apoptotic cellular death is crucial in its regulation to maintain the fertility potential of males [124].

In testicular germ cells of Cd-challenged animals, a trigger of apoptosis was observed via the endoplasmic reticulum stress-mediated apoptotic pathway, involving caspase-12 activation [125]. Apoptosis can also be induced via the mitochondrial pathway in germ cells with up-regulation of bcl-2-associated-X-protein (Bax), and down-regulation of B-cell lymphoma 2 (Bcl-2) genes. Although Bcl-2 inhibits apoptosis linking to the proapoptotic effector proteins [126], Bax modulates mitochondrial outer membrane permeabilization, thus causing apoptosis [53] and an increase in caspase-3 and -7. In fact, caspase-3 was demonstrated to be significantly high after 36 hours of exposure to Cd-treated rats, as also shown by Fouad et al. [58]. Instead, no significant variation of caspase 7 was observed [127].

Furthermore, a large number of TUNEL-positive germ cells, an increased level of Bax and a reduced level of Bcl-2 were observed in mice testes, thus confirming a negative role of CdCl_2_ on the seminiferous epithelium [44,47].

Therefore, these studies represent the complexity of different apoptotic pathways through which Cd induces apoptosis of testicular germ cells in different species that will help us in elucidating a common mechanism of apoptotic induction by Cd in testes [4].

Quercetin prevents apoptosis by modulating the expression of Bax, Bcl-extra-large (xL) and caspase-3 in germ cells [123]. similar results were obtained in Cd challenged rats protected with a grape seed proanthocyanidin extract (GSPE) [128]. Additionally, flavocoxid showed a similar activity against testicular damage induced by Cd [44].

Accordingly, we recently showed that BJe experimentally reduced apoptotic markers, thus protecting mice from Cd-induced testicular injury [47].

When rats exposed to Cd were treated with Cur, a phenolic compound obtained from Curcuma longa L. and largely used in traditional Asian medicine, reduced apoptosis, improved tubular morphology and testosterone levels were demonstrated [63].

Finally, our research lab experimentally investigated the protective and therapeutic effects of Re against CdCl_2_-induced toxicity in rat testes [46]. Indeed, Re showed a protective action against Cd testicular toxicity as it upregulated Bcl2 and downregulated p53 and Bax gene expression. Accordingly, a good improvement of functional and histopathological indicators was observed in the Re-pretreated mice [129].

## 7. Positive Effects of Nutraceuticals against Cadmium-Induced Testicular Injury on Blood–Testis Barrier Changes and Hormonal Imbalance

The BTB is formed by Sertoli cells by means of tight junctions [55]. In this way, the seminiferous epithelium is divided into a basal and an apical compartment, functionally and structurally different. A normal BTB is necessary to separate aploid germ cells from the immune system, and to prevent cytotoxic substances from passing from the circulation to the seminiferous epithelium [130]. When the BTB is damaged, germ cell loss and a reduced number of sperm occur with consequent subfertility or even infertility. After Cd challenge, BTB integrity is lost, owing to an evident increase in transforming growth factor (TGF)-β3, the most abundant form of TGF-β in the testis, produced by Sertoli cells, spermatogonia and early spermatocytes. TGF-β3 induces harmful effects in the tight junctions of Sertoli cells, owing to a decreased content in occludin, zonula occludens (ZO)-1, N-cadherin and claudin-11 [44,45,46,50]. In vitro studies showed that a correlation existed between tightness of the junctions and testosterone concentration [131].

In the testis, testosterone is required for the attachment of germ cells into the seminiferous tubules [132], and for the regular development of sperm cells [49]. Cd, besides causing a direct damage of the Leydig cells [46], is able to impair the hypothalamic-pituitary-gonadal axis [133], as demonstrated by the increased levels of follicle-stimulating hormone and luteinizing hormone, and by the decreased testosterone and inhibin-B serum levels [44].

We recently demonstrated that in mice treated with CdCl_2_, plus BJe alone or with associations with Cur-Re at different dosages, the seminiferous tubules had normal diameter and Johnsen’s scores, and the germinal epithelium showed mature spermatozoa [47].

Accordingly, in Cd-challenged mice, PDRN administration protected BTB ultrastructure, decreasing TGF-β3 immunoreactivity and enhancing its tight and adherens junctions [45].

Finally, MI alone or associated with Se significantly increased testosterone levels, ameliorated structural organization and increased the number of claudin-11 patches [46].

## 8. Conclusions

Cd is a highly diffused heavy metal and thus a ubiquitous environmental toxicant. The molecular mechanisms of Cd-induced structural and functional damages of the testicular tissue are a research topic of current interest. Dietary strategies and the use of nutraceuticals, especially, are recommended against Cd-induced testicular injury. Intriguingly, many studies indicate that the largest number of nutraceuticals has been found in foods typical of Mediterranean-style eating patterns, such as fresh fruits, beans, vegetables, grains and nuts [134,135,136]. Consequently, the addition in the diet of these compounds can be considered a new trustworthy medical tactic in subjects exposed to heavy metals, in particular to those whose mechanisms of action are similar to Cd (Table 1). Accordingly, a multifaceted mechanism of action by nutraceuticals appears crucial to counteract the detrimental molecular cascade in testicular injury caused by environmental heavy metals as Cd. Of course, we hope that the present narrative review, despite the limitations related to the type of review and lack of clinical data available to date, can be useful in improving the quality of life as well as environmental and food sustainability around the globe.

## Figures and Tables

**Figure 1 nutrients-14-00663-f001:**
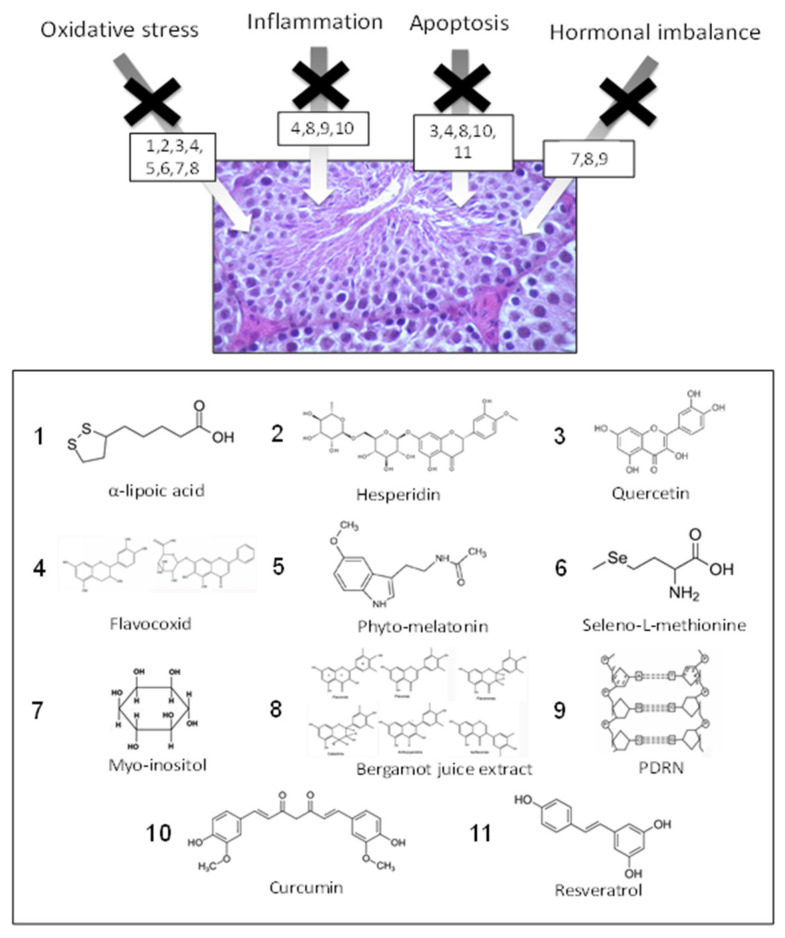
Structure of the nutraceuticals that are proposed to be effective against Cd-induced testicular injury. It is interesting to observe that in the majority of chemical structures here reported, multiple hydroxyl groups that typically confer a greater antioxidant (radical scavenging) capacity are present. This biochemical feature could functionally help to reduce inflammation and apoptosis, and modulate hormonal imbalance in Cd-induced testicular injury.

**Table 1 nutrients-14-00663-t001:** Table indicating the role of various nutraceuticals against Cd-induced testicular injury, including compounds, food sources, mechanisms of action and specific references.

Compound	Food Source	Mechanism of Action	References
α-lipoic acid	Spinach, broccoli	AO	[81,82,83]
Hesperidin	Citrus	AO–AI	[84,86]
Quercetin	Red apple, red onion, tomato	AO–AI	[123]
Flavocoxid	Cathechin (green tea, dark chocolate) Baicalin (onions)	AO–AI–AA–ABTBD	[44]
Phyto-melatonine	Coffe beans, apple, cherry, tomato	AO–AI	[91]
Seleno-L-methyonine	Brazilian nuts, potato, fish	AO–AHI	[93,94,96]
Myo-inositol	Cereals, citrus, dried plums, cantaloupe melon	AO	[46]
Bergamot juice	*Citrus bergamia* Risso et Poiteau (bergamot) fruits	AO–AI–AA	[47,113]
PDRN	Trout	AO–AI–AA–ABTBD	[45]
Curcumin	*Curcuma longa*	AO–AA	[47,63]
Resveratrol	Red wine, grape, peanuts, dried fruit	AO–AI–AA–ABTBD	[47]

AO = Antioxidant; AI = Anti-inflammatory; AA = Antiapoptotic; AHI = Antihormonal imbalance; ABTBD = Anti-blood–testis barrier damage.

## Data Availability

The datasets generated for this study are available on request to the corresponding author.

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
