# Peer review of "Nutraceuticals: A New Challenge against Cadmium-Induced Testicular Injury"

_nutrients, 2022, doi:10.3390/nu14030663_

Round 1

Reviewer 1 Report

The presented review is very interesting and, in my opinion. However, the manuscript is strongly underdeveloped and need significant improvements.

Below are my recommendations for improving the manuscript:

  1. First of all, there is a lack of lines numbering which makes it difficult to carry out a review.
  2. Introduction - According to my knowladge food is a primary route of exposure to cadmium for both, smokers and non-smokers, as with cigarettes c.a. 1 microg of Cd is inhaled (with one package) and with food 15-20 microg per day. Of course we can discuss that absorption from the respiratory tract is much higher than from gastrointestinal tract. However, since food is the main route of exposure, please provide a detailed information what levels of cadmium are consumed with food, and what differences can be observed between various regions or countries. For details please reffer to a paper: Dietary intake of metals by the young adult population of EasternPoland: Results from a market basket study. Journal of Trace Elements in Medicine and Biology 35 (2016) 36–42, http://dx.doi.org/10.1016/j.jtemb.2016.01.007
  3. Please provide also information on the amount which is inhaled with tobacco smoke and size of absorption from the respiratory tract.
  4. Introduction - there are various iso-types of metallothionein (MT), so we cannot say that there is only one. There are at least 4 types. Please describe them in relation to the cadmium detoxification mechanisms.
  5. Chapter CADMIUM AND TOXICITY OF TESTIS: POTENTIAL MOLECULAR TARGETS OF NUTRACEUTICALS - in this chapter please put at least basic information what are nutraceuticals, because all the described substances can be easily named "natural products", and it should be explained why why exactly nutraceuticals can be used against cadmium toxicity.
  6. Phytomelatonin - what is phytomelatonin? How it differs from "normal" melatonin? In which plants it may be found, and from which species it is obtained from? It must be better described.
  7. There is a need to put a separate Table which will summarize the role of various nutraceuticals  against cadmium-induced testicular injury, including compound, mechanism of action and foods in which these nutraceuticals can be found (particular food or at least source).

Author Response

Reviewer 1

We thank the Reviewer 1 for her/his suggestions aimed at improving the quality of our paper and at adding new, interesting topics.

Comments and Suggestions for Authors

The presented review is very interesting and, in my opinion. However, the manuscript is strongly underdeveloped and need significant improvements.

Below are my recommendations for improving the manuscript:

 1 - First of all, there is a lack of lines numbering which makes it difficult to carry out a review.

We are sorry for the evidenced problem, but we thought that during the process of manuscript submission the lines numbers would be included automatically.

2 - Introduction - According to my knowledge food is a primary route of exposure to cadmium for both, smokers and non-smokers, as with cigarettes c.a. 1 microg of Cd is inhaled (with one package) and with food 15-20 microg per day. Of course, we can discuss that absorption from the respiratory tract is much higher than from gastrointestinal tract. However, since food is the main route of exposure, please provide a detailed information what levels of cadmium are consumed with food, and what differences can be observed between various regions or countries. For details please reffer to a paper: Dietary intake of metals by the young adult population of Eastern Poland: Results from a market basket study. Journal of Trace Elements in Medicine and Biology 35 (2016) 36–42, http://dx.doi.org/10.1016/j.jtemb.2016.01.007

 As suggested by the Reviewer, we discussed the absorption of Cd from the gastrointestinal tract and inserted specific sentences to better clarify the levels of Cd present in foods and the possible differences observed among various regions or countries, as also indicated in the suggested reference, which was added in the list of the revised version of our manuscript.

3 - Please provide also information on the amount which is inhaled with tobacco smoke and size of absorption from the respiratory tract.

As requested by the Reviewer, data referred to the tobacco smoke and to the size of absorption through the respiratory tract were added in the first chapter of the review.

4 - Introduction - there are various iso-types of metallothionein (MT), so we cannot say that there is only one. There are at least 4 types. Please describe them in relation to the cadmium detoxification mechanisms.

As requested by the Reviewer, a more detailed description of the metallothioneins and the specific role of metallothionein-1 and -2 in heavy metal detoxification was added in the first chapter of the review.

5 - Chapter CADMIUM AND TOXICITY OF TESTIS: POTENTIAL MOLECULAR TARGETS OF NUTRACEUTICALS - in this chapter please put at least basic information what are nutraceuticals, because all the described substances can be easily named "natural products", and it should be explained why exactly nutraceuticals can be used against cadmium toxicity.

As suggested by the Reviewer, we added a new chapter entitled “NUTRACEUTICALS: THE NEW FRONTIER OF NUTRITION” where we put basic information what are nutraceuticals and why can be used against cadmium toxicity.

We added the following references:

- DeFelice S.L. The nutraceutical revolution: Its impact on food industry R&D. Trends Food Sci. Technol. 1995;6:59–61.

- Singh J., Sinha S. Classification, Regulatory Acts and Applications of Nutraceuticals for Health. Int. J. Pharma Biosci. 2012;2:177–187.

- Nasri H., Baradaran A., Shirzad H., Kopaei M.R. New concepts in nutraceuticals as alternative for pharmaceuticals. Int. J. Prev. Med. 2014;5:1487–1499.

6 - Phytomelatonin - what is phytomelatonin? How it differs from "normal" melatonin? In which plants it may be found, and from which species it is obtained from? It must be better described.

As suggested by the Reviewer, we better described the main characteristics of phytomelatonin. We added these informations in a revised version of manuscript:

“Structurally, melatonin and phytomelatonin are the same molecule. Specifically, melatonin is the term first proposed in 2004 and used to name to the compound of animal origin or obtained by chemical synthesis while the term phytomelatonin refers to melatonin of plant origin. Several studies have reported on detection of phytomelatonin in a variety of vegetables, fruits, seeds and medicinal herbs, and also in wild plants. To date, the phytomelatonin content of coffee beans is the highest recorded in plant material. Moreover, apple and cherry, Goji berry, tomato and pepper fruits also showed a high phytomelatonin content. Overall, aromatic and medicinal plants had higher levels of phytomelatonin than seeds and fleshy fruits, while leaves, stems, seedlings and roots presented higher phytomelatonin content than fruits.”

We added the following references:

- Blask D.E., Dauchy R.T., Sauer L.A., Krause J.A. Melatonin uptake and growth prevention in rat hepatoma 7288CTC in response to dietary melatonin: Melatonin receptor-mediated inhibition of tumor linoleic acid metabolism to the growth signaling molecule 13-hydroxyoctadecadienoic acid and the potential role of phytomelatonin. Carcinogenesis. 2004;25:951–960.

- Arnao M.B, Hernández-Ruiz J. The Potential of Phytomelatonin as a Nutraceutical. Molecules. 2018 Jan 22;23(1):238. doi: 10.3390/molecules23010238.

7 - There is a need to put a separate Table which will summarize the role of various nutraceuticals against cadmium-induced testicular injury, including compound, mechanism of action and foods in which these nutraceuticals can be found (particular food or at least source).

As suggested by the Reviewer, we put a separate Table which summarizes the origin, role, mechanism of action, and references of nutraceuticals discussed in the present review.

Reviewer 2 Report

General comment: Review article entitled “Nutraceuticals: A New Challenge Against Cadmium-Induced Testicular Injury” is an interesting study, with sufficient methodology and discussion of the results. Some minor corrections are required for the improvement of the manuscript.

Abstract: The Abstract is well written and adequately presents the aim and the basic results of the study.

-Could authors define the type of the review? Is this a systematic review? Could authors shortly describe the methodology used? eg key words, databases used, inclusion criteria of the studies selected.

Introduction: The introduction section is well-written and covers the basic aim of the study.

Materials and Methods:  The materials and methods are adequately presented.

-Could authors define the type of the review? Did authors use PRISMA guidelines or others?

-Authors should add a paragraph before introduction about the methodology used eg how many articles, inclusion criteria for the studies etc

Results: The results of the study are analytically presented.

-However authors could add 1 or 2 tables in order to shortly describe the basic studies, eg clinical studies, about the role of nutraceuticals.

Discussion: The results of study are sufficiently discussed.

-Authors could discuss possible limitations of the study.  

References: The references used by the authors cover adequately the relative scientific field and the aims of the study.

Author Response

Reviewer 2

General comment: Review article entitled “Nutraceuticals: A New Challenge Against Cadmium-Induced Testicular Injury” is an interesting study, with sufficient methodology and discussion of the results. Some minor corrections are required for the improvement of the manuscript.

We thank the Reviewer for his/her very favorable opinion on our work.

 Abstract: The Abstract is well written and adequately presents the aim and the basic results of the study. -Could authors define the type of the review? Is this a systematic review? Could authors shortly describe the methodology used? eg key words, databases used, inclusion criteria of the studies selected.

 As suggested by the Reviewer, we better described the methodology used for the present review.

Introduction: The introduction section is well-written and covers the basic aim of the study.

We thank the Reviewer for his/her positive comments.

Materials and Methods:  The materials and methods are adequately presented. - Could authors define the type of the review? Did authors use PRISMA guidelines or others? - Authors should add a paragraph before introduction about the methodology used eg how many articles, inclusion criteria for the studies etc

As suggested by the Reviewer, we better described the methodology used for the present review at the end of the Chapter Number 2 in the revised version of our manuscript.

Results: The results of the study are analytically presented. - However, authors could add 1 or 2 tables in order to shortly describe the basic studies, eg clinical studies, about the role of nutraceuticals.

As suggested by the Reviewer, we added a new chapter entitled “NUTRACEUTICALS: THE NEW FRONTIER OF NUTRITION”, where we put basic information what are nutraceuticals and why can be used against cadmium toxicity. Moreover, we added a separate Table which summarizes the origin, role and mechanism of action of nutraceuticals discussed in the present review.

Discussion: The results of study are sufficiently discussed. -Authors could discuss possible limitations of the study.

 As suggested by the Reviewer, we considered the possible limitations of the studies about this intriguing topic in the last paragraph of revised manuscript. Indeed, we hope that the present narrative review, despite the limitations related to type of review and lack of clinical data to date available, can be useful to improve the quality of life as well as the environmental and food sustainability around the globe.

References: The references used by the authors cover adequately the relative scientific field and the aims of the study.

We thank the Reviewer for his/her positive comments.

Round 2

Reviewer 1 Report

The Authors correctly answered to my questions and suggestions. The manuscript in the present form looks and sounds much more better. However, I still do not see the lines numbering. Moreover please correct as follows:

Conclusions: "and lack clinical data to date available" - this fragment does not sound good. Please revise it. 

Chapter 2. NUTRACEUTICALS: THE NEW FRONTIER OF NUTRITION - "organic acids (vitamin C)" - vitamin C is a lactone (cyclic carboxylic ester) and not a typical organic acid. Please revise.